# Unstable Coalescence Mechanism and Influencing Factors of Heterogeneous Oil Droplets

**DOI:** 10.3390/molecules29071582

**Published:** 2024-04-02

**Authors:** Zhuolun Li, Xiayi Huang, Xuenan Xu, Yujie Bai, Che Zou

**Affiliations:** 1Institute of Petroleum Engineering, Northeast Petroleum University, Daqing 163318, China; lzl_nepu@126.com (Z.L.); zc20180902@163.com (C.Z.); 2No. 9 Oil Production Plant of Daqing Oilfield Co., Ltd., Daqing 163318, China; huangxiayi@petrochina.com.cn; 3Drilling and Production Technology Research Institute, Petrochina Liaohe Oilfield Limited Company, Panjin 124010, China; xxuxuenan888@petrochina.com.cn

**Keywords:** emulsification, fluid flow disturbance, produced fluid, stability, surfactant

## Abstract

The use of a surfactant solution during oil and gas field development might improve the recovery rate of oil reservoirs. However, the serious emulsification of the produced liquid will bring challenges to the subsequent treatment process and storage and transportation. It is urgent to understand the coalescence mechanism of crude oil under the action of surfactant solution. This research investigates the coalescence mechanism of numerous oil droplets under liquid flow perturbation. The model was established to study the coalescence process of multiple oil droplets. The effects of the number of oil droplets under homogeneous conditions, the size of oil droplets, and the distance between oil droplets under non-homogeneous conditions on the coalescence process were analyzed. Meanwhile, the change rules of the completion time of oil droplet coalescence were drawn. The results show that the smaller the size of individual oil droplets under non-homogeneous conditions, the longer the coalescence completion time is, and when the size of individual oil droplets reaches the nanometer scale, the time for coalescence of oil droplets is dramatically prolonged. Compared to static circumstances, the time it takes for oil droplets to coalesce is somewhat shorter under gravity. In the fluid flow process, in the laminar flow zone, the coalescence time of oil droplets decreases with the increase of the liquid flow rate. However, in the turbulent flow zone, the coalescence time of oil droplets increases with the increase in the liquid flow rate. The coalescence time is in the range of 600~1000 ms in the flow rate of 0.05~0.2 m/s. In the presence of surfactants, the oil content in the emulsion system increases under the influence of pumping flow. The change in oil content rate with various surfactants is less impacted by flow rate, owing to the stable emulsion structure created by the extracted fluid within the reservoir. The study findings presented in this research provide technical assistance for effective crude oil storage and transportation.

## 1. Introduction

Although surfactant solution drive technology can effectively improve reservoir recovery [1], the recovery fluid produced from wells is heavily emulsified, creating certain difficulties for the ground treatment processes and storage [2]. In addition to crude oil and water, the extracted fluids in the reservoir and on the ground contained surfactants, clays, and other compounds, which can significantly impact the stability of oil droplets and alter the stability of the emulsion [3]. Numerous investigations have proven that both surfactants injected at the wellhead and surfactants produced by the reaction of alkali with crude oil in the reservoir significantly improve the stability of the recovered fluid [4,5]. In contrast, the synergistic action of alkali and surfactant produces an oil-in-water-type emulsion in the reservoir, which is substantially more stable [6]. The main reason for this increase in stability is that the substantial reduction in interfacial tension weakens the interfacial energy between oil and water, leading to easier mixing [7]. The stable emulsion can reduce the flow resistance. Although it is advantageous to flow underground, it will make the treatment of oil–water separation at the ground difficult [8,9]. Generally, oil–water separation is categorized into physical and chemical methods, among others [10]. Physical methods commonly used are gravity settling, centrifugal separation, and cyclone separation [11,12,13]. Adding the demulsifier directly into the emulsion is a chemical emulsion-breaking method, a proven technological tool for treating large quantities of extracted fluids [14,15]. Characterizing and evaluating the stability of emulsions can be obtained by measuring the height of the oil–water column [16]. The coalescence mechanism of droplets in oil and water phases plays a key role in the whole study process, whether during emulsification or emulsion breaking [17].

In the process of actual oilfield development and transportation, the stability and coalescing time of oil droplets are inevitably affected by the drag force of flooding displacement liquid systems during the flow of oil droplets in surface pipelines and porous media [18]. At the same time, oil droplets do not coalesce in pairs simultaneously, but multilevel and multidimensional oil droplets of different sizes coalesce simultaneously. The coalescing process has a deeper influence on the coalescing process of the whole system [19]. In general, the coalescence process of oil droplets is relatively complex and the coalescence process of oil droplets can be divided into the following two situations.

(1) Oil droplet coalescence process under static conditions

The coalescence process of oil droplets under static conditions is affected not only by the distance, density, viscosity, temperature, interfacial tension, and size of the droplets but also by the number of oil droplets under heterogeneous conditions and the relative position of the droplets.

(2) The oil droplet coalescence process under liquid flow disturbance

The coalescence process of oil droplets under liquid flow disturbance is mainly affected by gravity, flow rate, flow state, and pipe shape.

In previous studies, related scholars have investigated the mechanism of oil droplet coalescence in a steady state and divided the process of oil droplet coalescence into different stages, which are droplet convergence and liquid film evacuation, collision, rupture of interfacial film, and merging [20]. The effect of physical properties between oil and water (e.g., interfacial tension, droplet size, temperature, etc.) on droplet coalescence has been analyzed. The ions and pH value of the formation water also change the time of oil droplet coalescence [21]. More research has mostly investigated the separation effect of emulsions from the standpoint of the oil–water separation mode [22,23,24]. In our previous work, molecular dynamics simulation and microscope observation experiments studied the stability of different surfactant emulsions in the demulsification process and the microscopic mechanism of molecules in the oil film [25,26]. However, the microscopic mechanism of droplet coagulation had not been well described. The research outcomes of the droplet coagulation process in the presence of liquid flow disruption have not yet been explored. Consequently, the impact of drop number on oil droplet coalescence was investigated in this work using the level set computational technique and the finite element solution method to examine the oil droplet coalescence process. The influence of gravity and flow interference on oil droplet coalescence was studied, and the coalescence mechanism of the emulsion system in the wellbore was analyzed to provide technical support for efficient oil transport and storage.

## 2. Result Analysis

### 2.1. The Influence of the Number of Oil Droplets on the Coalescence of Oil Droplets

As illustrated in Figure 1, the spherical centers of the oil droplets are formed into equilateral triangles, quadrilaterals, and pentagons with side lengths of 20 mm in order to analyze the effects of the different numbers of oil droplets on the coalescence of the droplets. The radius of the oil droplets is 5 mm, the distance between the oil droplets is 20 mm, the density of the oil droplets is 800 kg/m^3^, the density of the water phase is 1000 kg/m^3^, the viscosity of the oil droplets is 3 mPa·s, and the viscosity of the water phase is 0.5 mPa·s. The color code in the figure shows the change in water phase saturation at the position of the liquid film.

The oil droplets are attracted to each other under the action of the van der Waals force, and coalescence occurs in the middle. The coalescence process of oil droplets shows that all oil droplets move closer to the center. The reason for this phenomenon is that the size and distance between oil droplets are exactly the same, the interfacial tension of liquid film between the oil phase and the water phase is the same, and the force between liquid film and liquid film is also completely the same, resulting in the phenomenon of multiple oil droplets moving to the middle at the same time in the process of coalescence. In order to analyze the influence of different numbers of oil droplets on coalescence, the time of coalescence and completion under different numbers of oil droplets was calculated, as shown in Figure 2.

It can be seen from Figure 2 that the coalescence time of oil droplets has no obvious change with different numbers of oil droplets. This is because the size of the simulated oil droplets and the distance between the oil droplets are the same, and the influence of fluid flow and gravity is not considered in the coalescence process, resulting in a strong similarity in the force and motion processes between the liquid films. Nevertheless, during the actual export of crude oil, oil droplets are not only affected by gravity and fluid carrying but also by the distance and size between oil droplets, resulting in a strong randomness in the coalescence process. Therefore, it is necessary to carry out further research on the coalescence process of oil droplets under heterogeneous conditions.

It can be seen from Figure 3 that when multiple oil droplets converge, the surface of the oil film has the same charge, so there will be a repulsion phenomenon between oil droplets during the coalescence process. Although there is a trend of convergence between oil droplets as a whole, the surface of the liquid film will move outside the oil film under the action of electrostatic repulsion, thus slowing down the convergence process. At the same time, comparing the process of three oil droplets and four oil droplets coalescing, it is found that when the oil film moves outward during the coalescing process, part of the oil film will be stretched so that the oil film will break and the oil droplets will finally coalesce.

### 2.2. Coalescence Process of Multiple Oil Droplets under Heterogeneous Conditions

Regardless of the stochastic nature of oil droplet movement and coalescence in formations and pipelines, most oil droplets coalesce in non-homogeneous environments. In non-homogeneous coalescence, the distance between oil droplets, droplet size, and other water phase parameters affect the entire process, resulting in a significant change in the coalescence pattern of oil droplets. Three oil droplets were selected to carry out the simulation, and the diameter of one of the droplets was changed to 3 mm, 4 mm, 5 mm, 6 mm, 7 mm, and 8 mm, respectively. The coalescence process of the oil droplets was simulated, as shown in Figure 4.

The coalescence process of the three oil droplets varies dramatically in response to changes in one of their sizes, and the coalescence completion time is significantly shortened. The coalescence time of the three oil droplets is 708.3 ms when their sizes are all 5 mm. The oil droplets did not coalesce to the center of the three oil droplets. Instead, they gradually coalesced to the location of the bigger diameter of the oil droplets. The coalescence duration of the oil droplets is 323.2 ms when the diameter of one of the oil droplets is adjusted to 8 mm. This phenomenon can be explained as the larger the size of the oil droplet is, the larger the van der Waals force at the position of the oil film is, and the stronger the attraction to the other two oil droplets is, thus reducing the coalescence time of the oil droplets. The state of oil droplets with different diameters after coalescence is further simulated.

As can be seen from Figure 5, taking three oil droplets as an example, after changing the size of one of the oil droplets, the state of the oil droplets after coalescence changed significantly. As the size of a single oil droplet among the three oil droplets increases, the position of the oil droplets after coalescence gradually decreases. This is due to the increase in the size of oil droplets, which enhances the adsorption capacity of other oil droplets. It illustrates that large oil droplets have a greater adsorption capacity than small oil droplets, and other oil droplets are close to the location of large oil droplets. Considering that the size of oil droplets also affects the coalescence time of oil droplets, according to the simulation results, the calculation completion time of oil droplet coalescence under different single oil droplet sizes is shown in Figure 6.

As one of the three oil droplets became larger, the completion time of oil droplet coalescence was reduced. The completion time of droplet coalescence was 950 ms for a 4 mm droplet and lowered to 300 ms when the droplet size was extended to 8 mm. The completion time of droplet coalescence was substantially lengthened when the size of a droplet approached the nanoscale scale. As a result, in order to increase emulsion stability, the size of oil droplets should be lowered for formation and coalescence in reservoirs. Furthermore, the emulsion has to coalesce as quickly as possible, aiming to prevent forming a smaller nanoscale emulsion. In addition to the size of the emulsion, the distance between the oil droplets in the emulsion contributes to its heterogeneous state in the reservoir. Consequently, the distance between the oil droplets and other places is altered, and the position of the third oil drop is shifted along the *y*-axis, with the original location serving as the origin. Figure 7 depicts the estimated process of oil droplet coalescence under various circumstances.

The location of the oil droplets significantly influences the coalescence shape of the oil droplets. The coalescence occurs preferentially between the two oil droplets and then further coalesces with the oil droplets at other locations. During this process, the third oil droplet gradually approaches and stretches during the coalescence of the other two oil droplets, and finally, the coalescence is completed. This indicates that the emulsion in porous media is due to its small space for oilfield development. In the absence of the action of surfactants, coalescence behavior between oil droplets is easily observed. In other words, although crude oil and water in porous media are easily emulsified and dispersed by the shear action of two-phase fluid, oil droplets are also prone to coalescence in narrow pore structures. The influence of oil droplet position on coalescence time is further studied, and the simulation results of different oil droplet positions on oil droplet coalescence completion time are provided.

As shown in Figure 8, the oil droplet location has a significant impact on the time it takes to complete coalescence. As the third oil droplet approaches the midpoint of the other two oil droplets, the time for oil droplet coalescence decreases, which can be as low as 400 ms, whereas the time for oil droplet coalescence increases exponentially with the distance between the third oil droplet and the midline of the other two oil droplets, which can be as high as 2500 ms. The coalescence process of oil droplets is actually a simultaneous process of multiple oil droplets, which are attracted to each other due to the fluid interface. In the non-homogeneous phase conditions, the distance between the oil droplets, size, and other factors affect the time of coalescence (for crude oil, the density of fluids is similar, generally in the range of 0.8~0.95 × 10^3^ kg/m^3^, while the viscosity and density have a certain correlation, the simulation process is relatively complex, so this study focuses on the analysis of the size and distance of oil droplets), so that there is a sequential relationship between their coalescence, which also causes the difference in coalescence time. Therefore, when the emulsion is subjected to oil–water separation, the small oil droplets can be prioritized to form large oil droplets by means of standing, which enhances its ability to coalesce other oil droplets. For emulsions with strong dispersion and high water content, small oil droplets must first complete the coalescence in a small space, and then the large oil droplets that have completed the coalescence enter a larger space to complete further coalescence.

### 2.3. The Coalescence Process of Oil Droplets under Fluid Flow Disturbance

In the above study, the focus was on the coalescence process of oil droplets under stable conditions without considering gravity. However, for surfactant-driven oil development, the crude oil in the reservoir is produced in the form of oil droplets. During the development process, the oil droplets are subjected to the trailing force of the water flow under all conditions except in the ground oil–water treatment devices such as settling tanks and dewatering units, and the coalescence process of the oil droplets is affected by the effect of the trailing force. Therefore, the coalescence process of oil droplets under the effect of gravity is analyzed.

As can be seen in Figure 9, the oil droplets move upward together while coalescing, eventually reaching the top to spread out and stabilize. Overall, this movement does not have an effect on the coalescence time of the oil droplets. In order to verify this idea, the variation of the coalescence completion time for different numbers of oil droplets subjected to gravity is calculated as shown in Figure 10.

The coalescence time of oil droplets does not change significantly under static conditions or gravity, but it does decrease under gravity. This is due to an increase in the speed of movement of oil droplets, which enhances the collision between oil droplets and the liquid film and thus reduces the coalescence time of oil droplets. As a result, it is determined that the relatively steady fluid flow can increase the coalescence speed of oil droplets. In the process of external transmission, in addition to the influence of gravity, the oil–water emulsion is also subjected to the trailing force of the liquid flow so that the oil–water emulsion undergoes further emulsification or emulsion breaking, thus changing the inlet flow rates of 0.005 m/s, 0.01 m/s, 0.02 m/s, 0.04 m/s, 0.08 m/s, 0.12 m/s, 0.18 m/s, and 0.2 m/s, respectively, and the change in the time of coalescence of the two oil droplets is analyzed. In order to analyze the effect of different flow rates on the completion time of oil droplet coalescence, flow rate variations can be classified into laminar and turbulent regions. The coalescence time curves of oil droplets at different flow rates are plotted.

Figure 11 demonstrates that in the laminar flow zone, the coalescence period of oil droplets reduces as the liquid flow rate increases. However, when the liquid flow rate increases in the turbulent zone, the time it takes for oil droplets to coalesce steadily increases. Because of the relative stability of the fluid flow in the laminar flow process, the oil droplets are subjected to the liquid flow’s carrying effect, increasing the likelihood of collision. At the same time, the liquid flow’s carrying kinetic energy supplies energy to breach the interfacial membrane of the oil droplets, reducing the time required for coalescence completion. From within the turbulent zone, it can be seen that the coalescence time of the oil droplets gradually increases with the increase in the liquid flow rate in the turbulent zone, which is completely opposite to the coalescence law of the oil droplets in the laminar flow stage. Under turbulent conditions, the fluctuation of the liquid flow is more violent, and the coalescence process between the oil droplets is more complicated. The simple study of two oil droplets affected by the disturbance of the liquid flow has some limitations. In order to investigate the coalescence and stability characteristics of multiple oil droplets further under different flow patterns, experimental studies on the coalescence process of oil droplets under liquid flow perturbation are carried out. The experimental results are shown in Figure 12.

The increase in oil content accelerates emulsification, and a portion of the water phase is disseminated in the oil, resulting in a water-in-oil emulsion. Figure 12 shows that when the pumping flow rate grows gradually, so does the oil content in the mixture, and the liquid flow rate is in the turbulence zone, but the space within the tube is significant, so the oil droplets in the emulsion system soon coalesce. Furthermore, the pumping flow rates and pumping flow rate curves show that during the fluid transportation process, it is primarily in a turbulent region; therefore, increasing the emulsion breakage rate of the emulsion by adjusting the flow rate of the fluid in the pipe is impossible. This issue must be considered when the replacement fluid injected into the formation has a substantial number of surface-active compounds (surfactants, alkalis, etc.) that might alter the interfacial tension between oil and water. A total of 0.3% sodium dodecylbenzene sulfonate, sodium dodecyl sulfate, petroleum sulfonate, betaine, and OP-10 surfactant are added to the mixture to compare the effects of various surfactants. In order to examine how various surfactants impact emulsification in relation to liquid flow, the tests produced variation curves for the pumping flow rate and the emulsion system’s final oil content under various surfactants, which are displayed in Figure 13.

After adding surfactants, the oil content in the emulsion system increased dramatically, with different surfactants causing varying increases. Overall, the pumping flow rate influences the rise in oil content in the emulsion system. As the pumping flow rate grows gradually, so does the oil content of the emulsion system, but the increase is limited. The curves representing the changes in oil content for various surfactants do not intersect or abruptly rise. This is because the extracted fluid in the reservoir’s porous media creates a stable emulsion structure, which prevents the emulsification level in the exterior and surface pipelines from changing noticeably.

## 3. Models and Methods

### 3.1. Basic Assumptions

Considering that the process of oil droplet coalescence and migration is complicated in the actual situation, the following assumptions are made to simplify the model.

(1)Brownian motion during oil droplet coalescence is not considered in the simulation;(2)The non-isothermal change in oil droplet coalescence and movement processes is not considered;(3)The influence of solid particles such as clay in formation on oil droplet coalescence is not considered;(4)The oil phase and water phase are homogeneous systems without considering the inhomogeneity of colloid asphaltene in the oil phase or the inhomogeneity of surfactant concentration in the water phase.

### 3.2. Oil Droplet Coalescence Model under Liquid Flow Disturbance

Due to the complexity of the actual situation, in order to fully consider the influence of different conditions on oil droplet coalescence, different mathematical models are set up for different situations.

(1) Fluid flow model ignoring inertia force and gravity
(1)$$\begin{array}{c} \rho \frac{\partial u}{\partial t}=\nabla \cdot \left[-pI+K\right]+F\\ \rho \nabla \cdot u=0\\ K=\mu \left(\nabla u+{\left(\nabla u\right)}^{T}\right) \end{array}$$

Here, *ρ* is the density, kg/m^3^, *μ* is the dynamic viscosity, N/m^2^, *u* is the velocity, m/s, *p* is the pressure, Pa, *g* is the gravity vector, m/s^2^, and *F* is the surface tension force acting at the air/water interface.

(2) Fluid flow model that ignores inertia force and considers gravity
(2)$$\rho \frac{\partial u}{\partial t}=\nabla \cdot \left[-pI+K\right]+F+\rho g$$

(3) Laminar flow model considering gravity
(3)$$\rho \frac{\partial u}{\partial t}+\rho \left(u\cdot \nabla \right)u=\nabla \cdot \left[-pI+K\right]+F+\rho g$$

(4) Two-phase flow level set model
(4)$$\frac{\partial \varphi }{\partial t}+u\cdot \nabla \varphi =\gamma \nabla \cdot \left(\varepsilon_{ls}\nabla \varphi -\varphi \left(1-\varphi \right)\frac{\nabla \varphi }{\left|\nabla \varphi \right|}\right),\varphi =phils$$

Due to the small velocity of liquid flow under an electric field, according to Stern’s theory of the double layer, the double layer can be divided into two parts using a plane called the Stern plane (actually an imaginary plane): the inner Stern layer and the outer diffusion layer. The Stern plane is about the radius of the hydrated ion from the surface of the solid. It is an imaginary surface formed by the connection of the adsorbed ion center and the formula of the repulsion force of the double electric layer when two droplets are close to each other.
(5)$$P_{v}^{*}=D^{*}\exp\left(-\kappa^{*}h^{*}\right)$$
(6)$$D^{*}=64n^{*}k^{*}T^{*}\tanh^{2}\left(\frac{Z\varphi^{*}e^{*}}{4k^{*}T^{*}}\right)$$
(7)$$\kappa^{*}={\left(\frac{8\pi n^{*}Z^{2}e^{*2}}{\varepsilon k^{*}T^{*}}\right)}^{1/2}$$

Here, *e** is the unit charge, *ε* is the dielectric constant, *κ* is the Debye length, *k** is the Boltzmann constant number, *T** is the absolute temperature, *n** is the ion concentration, *z* is the valence number of ions, and *ϕ** is surface potential.

The quantitative characterization of heterogeneous oil droplet coalescence was realized through the above theory, combined with the COMSOL finite element simulation method.

### 3.3. Experimental Part

The oil–water two-phase flow pipe flow experiment adopts a self-built experimental device, which was built by the College of Petroleum Engineering at Northeast Petroleum University. Figure 14 is the schematic diagram of the experimental device.

The composition of the experimental device includes a liquid supply system (including a stirring device), a temperature control system, and a parameter measurement system. The liquid supply system was used to pump the prepared solution into the pipeline and adjust the flow rate through the valve. The agitator was used to stir the solution to stratify the water–crude mixture. The temperature control system was used to keep track of the temperature of the sample, including the temperature controller, heating device, cooling device, and measuring point heat tracing system. The parameter measurement system was mainly used to measure the flow rate of the fluid passing through the vertical experimental pipeline under stable flow conditions and the pressure difference at the corresponding point, which was read from the pressure sensor.

The experimental steps are as follows:(1)The parameterization system was turned on to ensure that the initial readings of the meters were correct;(2)The water level in the reservoir was controlled at the required level, and the temperature control system was turned on;(3)After combining 0.5 m^3^ of crude oil and 1 m^3^ of water in the supply tank, the oil–water combination was thoroughly emulsified for 30 min at a shear flow rate of 3000 r/min using an emulsifying stirrer. To enable the fluid supply to reach the experimental displacement, the fluid supply system was turned on, and the output valve was adjusted;(4)Various injection displacements were used, including 0.02 m^3^/min, 0.05 m^3^/min, 0.1 m^3^/min, 0.2 m^3^/min, and 0.5 m^3^/min. The oil–water emulsion was circulated for 30 min after the fluid flow inside the experimental pipeline stabilized. The oil–water mixture in the cycle was quickly taken out and centrifuged in a test tube. The centrifugal speed was 3000 r/min, and the centrifugal time was 10 min. After centrifugation, the samples were taken out, and the volume of the oil phase and water phase separated from the mixture was read. The demulsification rate of the emulsion was calculated, and the effects of different flow rates on the stability of the oil–water emulsion were compared and analyzed,(5)The liquid in the experimental line was drained, and the experimental equipment was shut down.

## 4. Conclusions

In this study, the coalescence mechanism of multiple oil droplets under liquid flow disturbance conditions was investigated. Upon analysis, the coalescence process of oil droplets in non-homogeneous phase circumstances changes significantly, and with the increase in droplet size, coalescence time in the process of oil droplet aggregation decreases. When individual oil droplets reach nanoscale size, it takes significantly longer for them to combine. The oil droplet coalescence time decreases slightly under gravity compared to static conditions. During the fluid flow process, the coalescence time of oil droplets decreases with the increase in liquid flow rate in the laminar flow region but increases with the increase in liquid flow rate in the turbulent flow region. In the event of an addition of a surfactant, the oil content in the emulsion system rises and is influenced by the pumping flow rate.

## Figures and Tables

**Figure 1 molecules-29-01582-f001:**
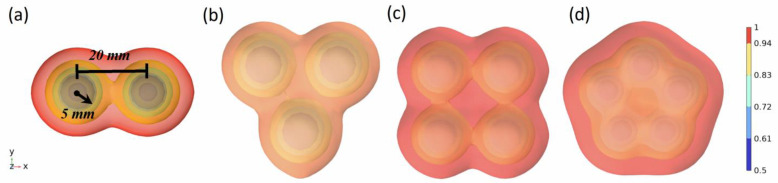
The influence of different numbers of oil droplets on coalescence (oil phase volume fraction distribution). (**a**–**d**) is the initial state of coalescence of 2, 3, 4, and 5 oil droplets, respectively.

**Figure 2 molecules-29-01582-f002:**
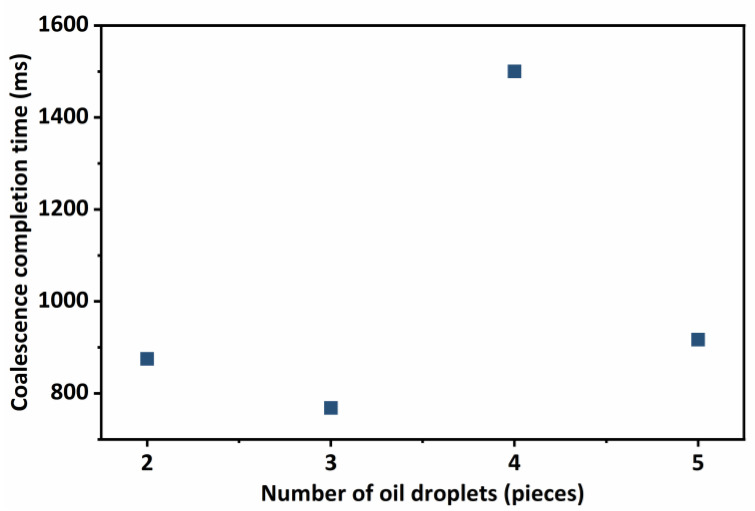
The coalescence time of oil droplets under different numbers of oil droplets.

**Figure 3 molecules-29-01582-f003:**
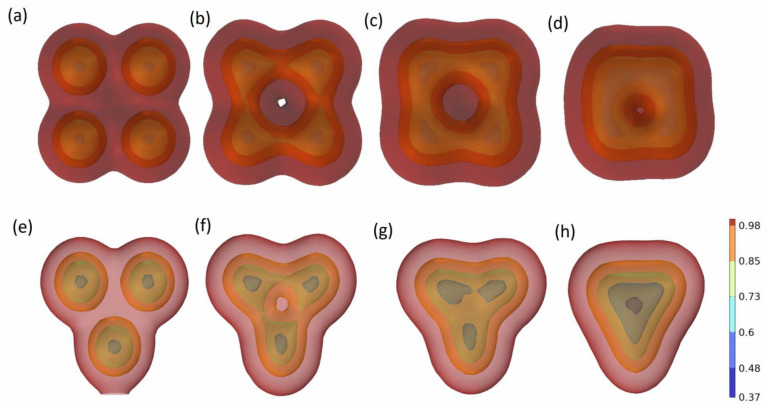
The coalescence process of multiple oil droplets under the action of electrostatic force. (**a**–**d**), (**e**–**h**) are respectively the coalescence processes of four and three oil droplets at 0 ms, 100 ms, 200 ms, and 500 ms, respectively.

**Figure 4 molecules-29-01582-f004:**
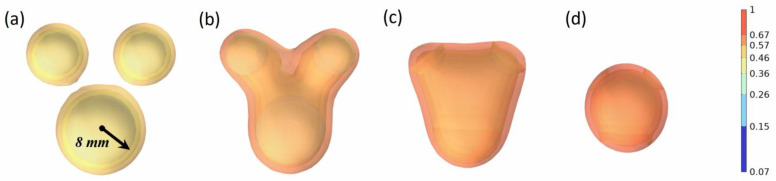
The coalescence process of three oil droplets when the diameter of one oil droplet is 8 mm (oil phase volume fraction distribution). (**a**–**d**) indicate the coalescence states at times of 0 ms, 100 ms, 200 ms, and 400 ms, respectively.

**Figure 5 molecules-29-01582-f005:**
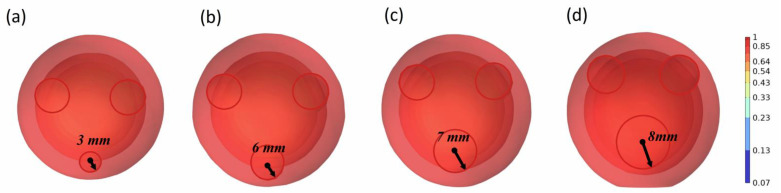
State of oil droplets after coalescing (oil phase volume fraction distribution). (**a**–**d**) indicate the coalescence state of the third oil droplet for 3 mm, 6 mm, 7 mm, and 8 mm conditions, respectively.

**Figure 6 molecules-29-01582-f006:**
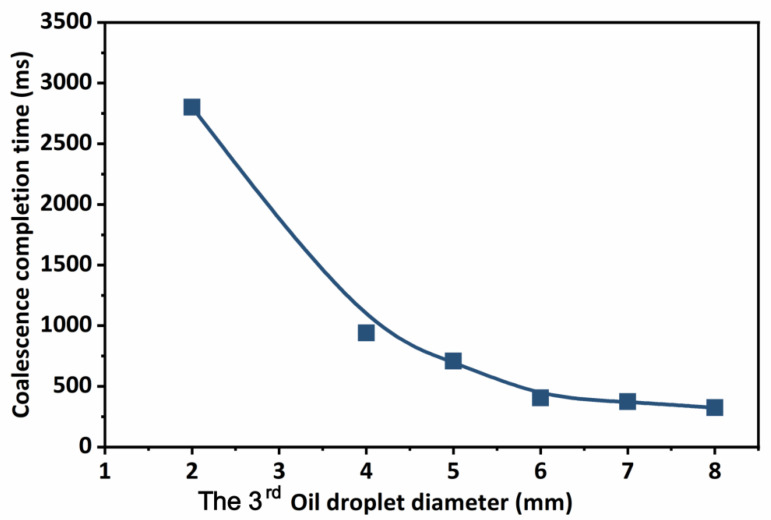
Time for oil droplet coalescence to complete under different single oil droplet sizes.

**Figure 7 molecules-29-01582-f007:**
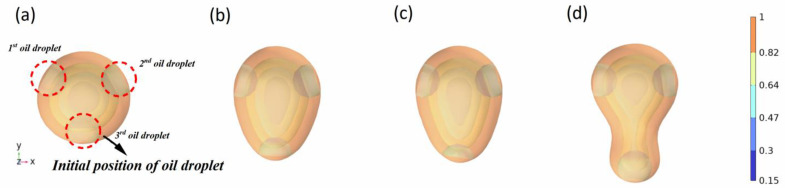
Influence of oil droplet position on oil droplet coalescence morphology (oil phase volume fraction distribution). (**a**–**d**) are the coalescence states of the third oil droplet in the *y*-axis direction located at 2 mm, −4 mm, −6 mm, and −8 mm for a time of 1000 ms.

**Figure 8 molecules-29-01582-f008:**
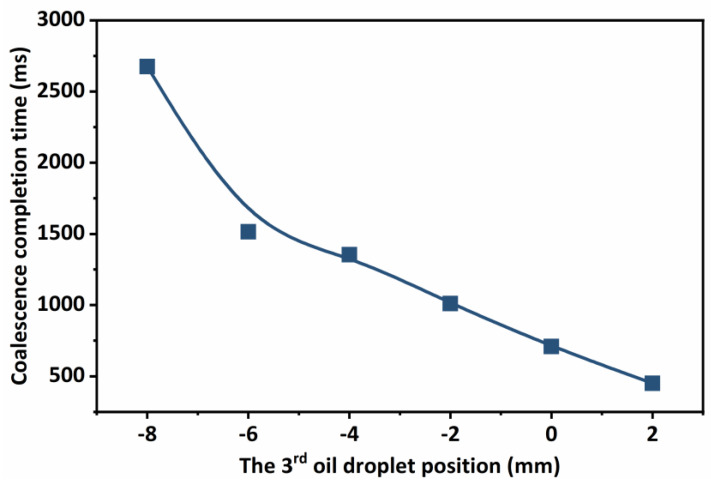
Influence of oil droplet position on coalescence time of oil droplets.

**Figure 9 molecules-29-01582-f009:**
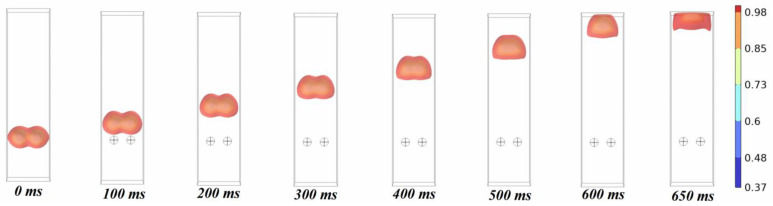
The coalescence process of oil droplets under the action of gravity (oil phase volume fraction distribution). “⨁” represents the initial position of the oil droplet.

**Figure 10 molecules-29-01582-f010:**
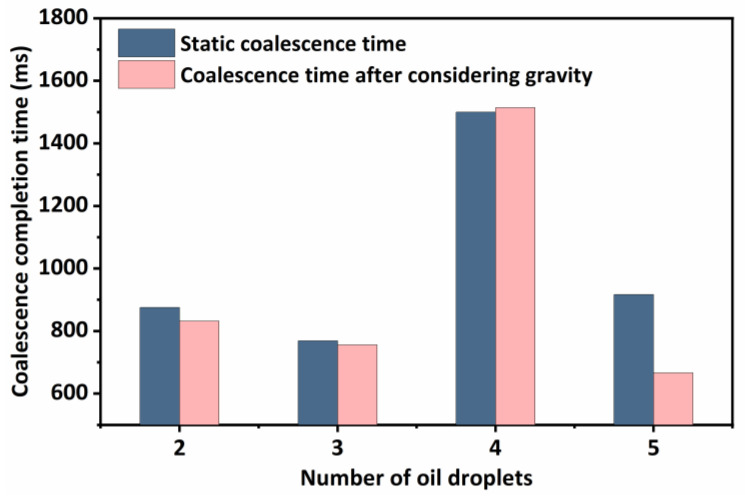
Comparison of oil droplets coalescence and completion time.

**Figure 11 molecules-29-01582-f011:**
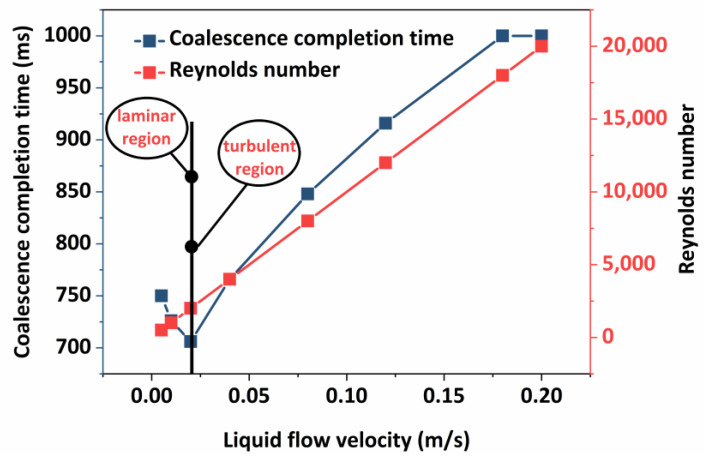
The coalescence time of oil droplets at different flow rates.

**Figure 12 molecules-29-01582-f012:**
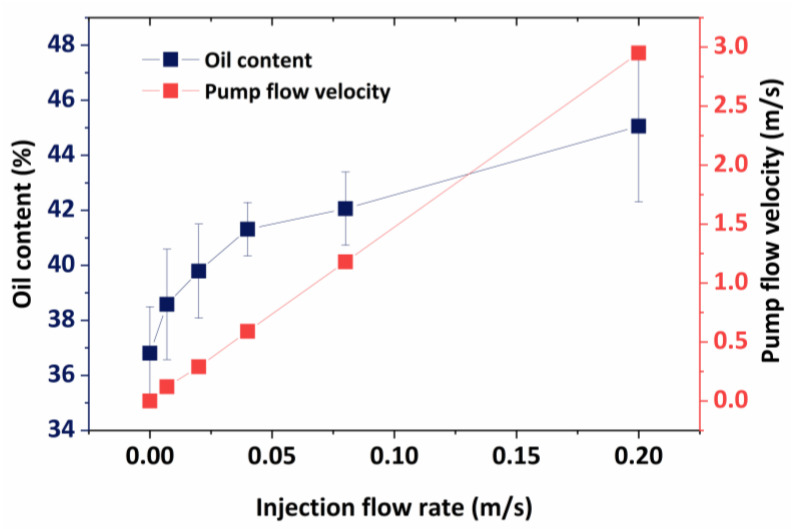
Changes in the oil content of the mixture under different pumping flow rates.

**Figure 13 molecules-29-01582-f013:**
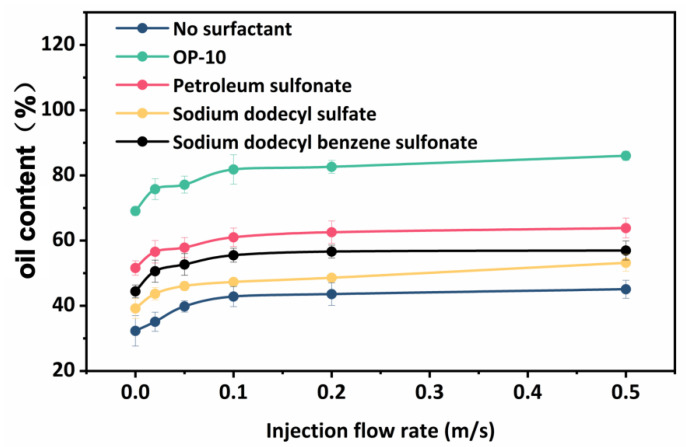
Variation curve of pump flow rate and oil content of emulsion under different surfactants.

**Figure 14 molecules-29-01582-f014:**
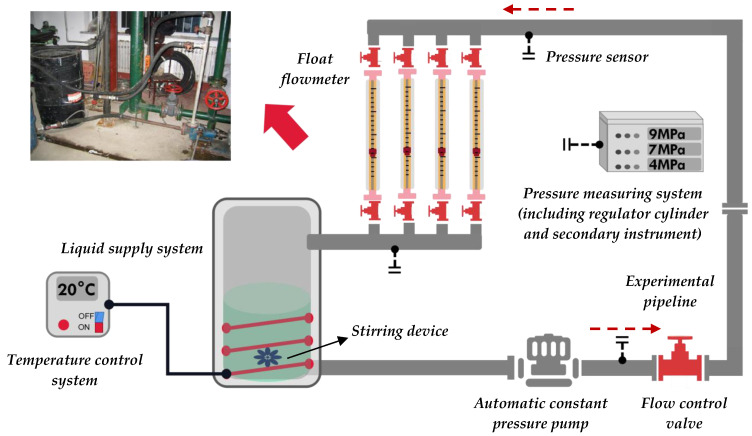
Schematic diagram of simulating experimental device.

## Data Availability

Data are contained within the article.

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
