# Peer review of "Unstable Coalescence Mechanism and Influencing Factors of Heterogeneous Oil Droplets"

_molecules, 2024, doi:10.3390/molecules29071582_

Round 1

Reviewer 1 Report (New Reviewer)

Comments and Suggestions for Authors

In this paper, the coalescence mechanism of multiple oil droplets under liquid flow disturbance conditions was investigated. Upon analysis, the coalescence process of oil droplets in non-homogeneous phase circumstances changes significantly, and increasing droplet size reduces coalescence time during the aggregation of many oil droplets.

The layout of the article is confusing, the quality of English is poor, and many non-academic expressions appear, so it will be reconsidered after major revision.

To improve the manuscript, the authors are suggested to consider the following comments.

1.     The part of Introduction is not comprehensive enough, it is suggested to add related content,

2.     Figure 1 is a bit incomprehensible and it is recommended to be redrawn or deleted,

3.     Figure 3, it is better to change the histogram to a scatter chart,

4.     Figure 7, the x-axis annotation is incorrect, change ‘drop’ to ‘droplet’,

5.     The unit of flow rate is ‘m3/s’, and the unit of velocity is ‘m/s’, they were confused in this manuscript,

6.     I have marked the other problems of writing in yellow in the manuscript.

Comments on the Quality of English Language

the quality of English is poor, and many non-academic expressions appear

Author Response

Reviewer 2 Report (New Reviewer)

Comments and Suggestions for Authors

The coalescence mechanism of multiple oil droplets under fluid flow disturbance conditions has been investigated. The contents of this paper are of interest to researchers. However, the manuscript has some shortcomings and cannot be considered for publication in its present form. The following points are raised by the reviewer as questions and concerns.

1.     Are the level set calculation techniques and finite element studies based on ANSYS or COMSOL? What software was used to simulate the results obtained in Figures 2, 4, 5, 6, 8, 10, etc.? This must be clearly stated in the "Modelling and Methods" section.

2.     Fig. 12, Why does the coalescence completion time have a minimum value near a liquid flow velocity of 0.025?  “Figure 12 demonstrates that in the laminar flow zone, the coalescence period of oil droplets reduces as the liquid flow rate increases.” This statement is not consistent with the graph.

3.  Some experimental results do not have error terms, such as Fig. 13,14 etc.

4.     How the oil content is determined must be clearly stated in the "Modelling and Methods" section.

5.     “Furthermore, the pumping flow rate and pumping flow rate curves show that during the fluid transportation process, it is primarily in a turbulent region, so increasing the emulsion breakage rate of the emulsion by adjusting the flow rate of the fluid in the pipe is impossible.” Whether it is in the turbulent region must to be determined by the Reynolds number, not the flow rate or flow volume.

Don't the pump flow and pump flow curve (or Figure 13, injection flow and pump flow velocity) have the same meaning when determining the pipeline?

The liquid flow velocity in Fig 12 is an order of magnitude lower than the pump flow velocity in Figure 13, why?

6.     Conclusion section “The change in oil content rate with different surfactants is less affected by flow rate, which is owing to the stabilized emulsion structure created by the extracted fluid within the porous medium”. However, the description in the "Modelling and methods" section indicates that no flow experiments were carried out in porous media.

Author Response

Reviewer 3 Report (New Reviewer)

Comments and Suggestions for Authors

In the current paper the authors discuss a coalescence mechanism of oil droplets under different conditions in homo- and heterogeneous environments. Authors presented the work in an effective manner using different model structures that help the reader to understand easily. I recommend publishing this work after a few minor comments:    

 In this paper there are many sections that need to be improved. See the comments:

1.      Some of the Figure’s resolution can be improved for better understanding, especially Figure 6, the 7 mm and 8 mm droplets are hardly visible.

2.      I recommend authors to discuss more about coalescence process using different heterogeneous state and its conditions, I mean how another heterogeneous state affects the coalescence?

3.      Authors should explain the color bar somewhere in main text or supporting information.

4.      I was just wondering why authors didn’t include 3rd droplet in Figure 7.

Based on everything mentioned above, I would recommend publication after major revisions.

Round 2

Reviewer 1 Report (New Reviewer)

Comments and Suggestions for Authors

The authors addressed all the comments.

Comments on the Quality of English Language

The authors addressed all the comments.

Author Response

Dear editor, we have sorted out the language problems in the paper again according to your suggestions, and changed some language and format problems.

Reviewer 2 Report (New Reviewer)

Comments and Suggestions for Authors

The author has revised and explained the issues raised by the reviewer, but one question remains. Don't "injection flow rate" and "pump flow rate" mean the same thing? But why is there such a big difference in the values of "Injection flow rate" and "Pump flow rate" in Figure 13?

Author Response

Dear editor

The flow rate in Figure 13 is the same as the previous flow rate, but different from the previous experiment, we added a large flow experiment at the end of the experiment, increasing the flow rate to 0.5m/s, and analyzed whether continued increase of the flow rate would have an impact, so an additional experimental result was added at the end.

This manuscript is a resubmission of an earlier submission. The following is a list of the peer review reports and author responses from that submission.

Round 1

Reviewer 1 Report

Comments and Suggestions for Authors

In this manuscript, the authors have developed a simple model for oil droplet coalescence and investigated the impact of factors such as oil droplet number and particle size on the coalescing time of oil droplets during transportation. However, due to a lack of innovation and some existing problems in this research, I do not recommend the publication of this article.

1、The introduction of this paper is just a simple reference to the previous research work, which lacks logic. In addition, part of the content has nothing to do with this research, such as the reference relating to solid particles and scaling.

2、The established model is too simple to study the process of oil droplet coalescence. In Page1, there is not the basic mathematical models which describing the oil droplets interaction.

3、In this article, it is unreasonable to use the van der Waals force as the main driving force of the coalescence between oil droplets under such a macroscopic condition.

4、When considering the size and number factors that affect the coalescence time of oil droplets, the experimental data used are too few. Especially, in Figure 5, the obtained experimental data cannot support the experimental conclusion.

5、Page 11 line 12, here is a punctuation error. Figure 3 lacks a legend.

Comments on the Quality of English Language

Extensive editing of English language required

Reviewer 2 Report

Comments and Suggestions for Authors

Dear Authors,

Kindly find my whole comments and modification in the attached PDF file

Best luck

Comments on the Quality of English Language

Moderate English modification is required.

Best luck

Reviewer 3 Report

Comments and Suggestions for Authors

The aim of the authors in this article was to analyze the mechanism of coalescence of several oil droplets in ternary production fluids of polymer, alkali, and surfactant under fluid flow disturbance, that the effect of the number of oil droplets in homogeneous conditions, the size of oil droplets, and the distance between oil droplets on the coalescence of oil droplets in Heterogeneous conditions have been analyzed. To simplify the model, the authors have considered some assumptions that need to be explained. What is the reason for choosing these factors? When you ignore Brownian motion, does it not affect the final results? Also, the experiments were carried out on a self-made experimental device built by the Faculty of Petroleum Engineering, Northeast Petroleum University. How reliable is the data extracted from this device? Is proper calibration done on the device? Also, the standard writing instructions in the text of the article have not been followed. The English language of the article needs to be improved.

There are also several comments that should be corrected next to the main question.

Abstract:

State the quantitative results in the abstract. The authors have only presented the results in general.

-           The results show that the coalescence time of oil droplets …”. The results Showed is correct.

Keywords:

-          Sort the keywords alphabetically.

Models and Methods:

-          In the experimental part, you have described the method of conducting the experiment as if you were explaining the user manual of the device to the reader. Express the sentences based on your experiments.

Conclusion:

-          At the beginning of the conclusion, briefly state the aim of the research

Round 2

Reviewer 2 Report

Comments and Suggestions for Authors

The present form of the recent MS is fulfill the requirements for publication in journal of processes...

Comments on the Quality of English Language

Minor editing of English language required

Reviewer 3 Report

Comments and Suggestions for Authors

Dear authors

I wish you success in your future researches.